# Factors Associated with the Uptake of Rotavirus and Pneumococcal Conjugate Vaccines among Children in Armenia: Implications for Future New Vaccine Introductions

**DOI:** 10.3390/vaccines11111719

**Published:** 2023-11-15

**Authors:** Anya Agopian, Heather Young, Scott Quinlan, Madeline Murguia Rice

**Affiliations:** Department of Epidemiology, Milken Institute School of Public Health, George Washington University, 950 New Hampshire Ave NW 5th Floor, Washington, DC 20052, USA; youngh@gwu.edu (H.Y.); squinlan@email.gwu.edu (S.Q.); mrice@bsc.gwu.edu (M.M.R.)

**Keywords:** vaccinations, childhood vaccinations, childhood immunizations, new vaccines, Armenia

## Abstract

Advances in vaccinology have resulted in various new vaccines being introduced into recommended immunization schedules. Armenia introduced the rotavirus vaccine (RV) and the pneumococcal conjugate vaccine (PCV) into its national schedule in 2012 and 2014, respectively. Using data from the Armenia Demographic and Health Survey, the uptake of the RV and the PCV among children aged younger than three years was estimated. Multilevel logistic regression models were used to evaluate individual- and community-level factors associated with uptake. Intra-cluster correlations were estimated to explain variations in uptake between clusters. The uptake proportionof each RV dose were 90.0% and 86.6%, while each PCV dose had values of 83.5%, 79.4%, and 75.5%, respectively. Non-uptake was highest among children less than 6 months old, children with one sibling, children from a wealthy family, or children whose living distance to a health clinic was problematic. Significant variability in non-uptake due to cluster differences was found for both RV doses (30.5% and 22.8%, respectively) and for the second PCV dose (53.9%). When developing strategies for new vaccine implementation, characteristics of the child, such as age, siblingship, and distance to a health clinic or residence, should be considered. Further exploration of cluster differences may provide insights based on the increased uptake of these and other new vaccines.

## 1. Introduction

New vaccine introductions (NVIs) into recommended immunization schedules have increased in recent years due to advances in vaccinology, further expanding the scope of vaccine-preventable diseases and, in turn, decreasing morbidity and mortality rates attributable to these diseases. NVIs have been shown to reduce disease burden, not only among those vaccinated, but also in the overall population [1,2,3]. In order to ensure a proper level of protection and reduce disease burden in the community, examining the initial implementation and uptake of these NVIs is crucial in informing future vaccination activities.

Although national immunization programs, health systems, and disease burdens differ between countries, there are several factors that have been found to be commonly associated with NVIs. Immunization programs have been shown to have a positive effect on health systems due to increased technical assistance and an anticipated decrease in disease burden [4,5,6,7]. Unfortunately, NVIs have also been observed to have a negative impact on health systems as a result of inadequate planning and, consequently, logistical issues and vaccine shortages [7]. A proper assessment of NVIs should include examining factors associated with uptake to identify potential areas for interventions in order to increase coverage. Previously established factors include structural issues related to immunization programs [8,9,10], residence [11,12,13,14,15], socioeconomic status [13,16,17,18], parental education [14,16], and siblingship [12,16,18,19,20]. The COVID-19 pandemic has greatly impacted individuals’ wellbeing and further revealed the importance of understanding drivers of vaccine uptake. Although the factors that have thus far been identified with the uptake of a COVID-19 vaccine have varied, the pandemic has demonstrated that vaccine uptake is a nuanced area of concern that must be regularly examined to ensure strong uptake and proper protection in the community [21,22,23,24].

With the support of Gavi, as part of the Vaccine Alliance, a global health partnership aimed at increasing access to immunizations in poor countries, the rotavirus vaccine (RV) was introduced in Armenia in 2012, with a recommendation of two doses delivered at 6 and 12 weeks of age. Initial evaluative studies found that, in the two years following the introduction of the RV, reported coverage was 90.4% for the first dose and 91.3% for the second dose [25]. Studies have also shown that the introduction of the RV in Armenia drastically reduced rotavirus-related hospitalizations in children under one year of age, as well as across all age groups, presumably due to conferred herd immunity [26]. Armenia also introduced a pneumococcal conjugate vaccine (PCV) in 2014 with a recommendation of three doses delivered at 6, 12, and 18 weeks of age [25]. The uptake and potential impact of the PCV among children living in Armenia have yet to be examined.

To better plan for immunization programs and further reduce the burdens of rotavirus and pneumococcal diseases in Armenia, our objectives were to (1) quantify uptake of the RV and the PCV among children living in Armenia in 2015–2016 and (2) examine the individual- and community-level factors associated with the non-uptake of each dose of the vaccines. Accomplishing these objectives may identify areas of improvement and foci for interventions.

## 2. Materials and Methods

The Armenian Demographic and Health Survey (ADHS) is a serial, nationally representative sample survey that has been conducted by the National Statistical Service and the Ministry of Health of Armenia since 2000, and is part of MEASURE DHS, a worldwide USAID-sponsored project. The 2015–2016 ADHS implemented a two-stage sample design to reach the target sample sizes [27]. In the first stage, clusters were selected from a list of enumeration areas covering Armenia that was provided by the National Statistical Service of Armenia. In the second stage, a complete listing of households in each selected cluster provided the sampling frame from which households were chosen. Women from these selected households were eligible if they were between the ages of 15 and 49 years and if they were either permanent residents or visitors from the night before. Those eligible provided verbal informed consent to participate in the svey.

This project was a secondary analysis of an existing de-identified data set from the ADHS and was determined by The George Washington University Institutional Review Board to not be human research. The analytic sample for this analysis consisted of all living children aged 0 to 35 months with documented birthdates. The subsamples for each vaccination were composed of children eligible for that particular vaccination and dose based on the recommendations of Armenia’s National Immunization Program schedule and the year the vaccine was introduced.

Vaccination information was collected for all children under three years of age in each household. Documentation of immunization was obtained either from child health cards, which were maintained by local health facilities, or immunization passports, which were kept by the child’s parent or guardian. Data were collected from both sources if available. If neither was available, vaccination history reported by the mother was recorded. If there was no vaccination documentation and the mother could not recall any details, the vaccination was decidedly not administered, as per ADHS documentation.

Uptake, defined as the documented receipt of each of the doses of the RV and the PCV, was assessed by determining the weighted proportion of children who had received each dose. Frequencies, proportions, and 95% confidence intervals (CIs) for the estimates of uptake of each dose of the RV and the PCV were calculated.

Multivariable multilevel logistic regression models were used to examine the individual and contextual factors associated with RV and PCV non-uptake, defined as those without a documented date of vaccination, among children with vaccination cards and those eligible for the particular vaccination. Potential factors, based on a literature review, included individual-level factors such as the child’s sex, age, siblingship, and birth order, as well as the mother’s age, education, employment status, antenatal care, and place of delivery, and the household’s wealth index. Contextual or community-level factors, such as distance to health clinic (categorized as “not a big problem” and “big problem” by the ADHS), place of residence, and region, were also considered. Multicollinearity was assessed by examining the variance inflation factor (VIF) using less than five as criteria to rule out multicollinearity. Birth order and siblingship were found to be multicollinear, resulting in birth order being excluded from subsequent analyses. Wealth index, a composite measure created by the ADHS using principal component analysis, was based on a household’s ownership of certain assets, housing construction materials, and access to water and sanitation. For the purposes of this analysis, wealth index was further grouped into three categories: poorer/poorest as low, middle as middle, and richer/richest as high. Armavir, due to its size and composition as the largest province in Armenia with both rural and urban regions, was the reference for these analyses.

Prior to determining the final multilevel multivariable model, individual-level factors and community-level factors were evaluated in separate models. The final multivariable multilevel logistic regression model included factors that were significant in the individual- and community-level models. Fixed effects were assessed by the adjusted odds ratios (aORs) with 95% CIs. The intra-cluster correlations (ICCs) were estimated to explain the measures of variation between clusters (random effects) based on a null model and each of the aforementioned multivariable models. ICCs were calculated using the following formula: τ_00_/[τ_00_ + (π^2^/3)], where τ_00_ is the covariance parameter estimate generated from the model. The fit of each of the models was assessed by examining the log likelihood and Akaike information criterion (AIC) statistics. Based on the literature [18], potential interactions between the wealth index and residence were assessed, and a significant interaction was found in the models for the RV; thus, stratified analyses were conducted.

All tests were two-sided and *p* < 0.05 was used to define statistical significance. Due to the study design of the ADHS, all statistical methods utilized techniques and included weights where possible to account for the complex sampling design. All analyses were performed using SAS version 9.4 (SAS Institute Inc., Cary, NC, USA). No imputation for missing data was performed.

## 3. Results

In accordance with the introduction of each of the vaccines, the ADHS had information on 1017 children born after 2012 for the analysis of RV uptake and 371 children born after 2014 for the analysis of PCV uptake. Demographic and other select characteristics of the children and their mothers based on vaccine type are presented in Table 1. 

Health cards were available for 93.3% of children eligible for the RV and 93.0% of children eligible for the PCV. Table 2 displays the vaccination information for all living children under three years of age based on vaccination type. Among the children eligible for the RV, 90.0% had received the first dose (RV1) and 86.6% had received the second dose (RV2). Among the children eligible for the PCV, 83.5% had received the first dose (PCV1), 79.4% had received the second dose (PCV2), and 57.5% had received the third dose (PCV3).

Table 3 presents the final multilevel models for both RV doses stratified by urban–rural residence. Among the children living in urban areas, lower odds of RV1 non-uptake were observed in older children when compared with children aged 0 to 5 months (6 to 11 months: aOR = 0.34, 95%CI: 0.14–0.84; 12 to 17 months: aOR = 0.18, 95%CI: 0.04–0.77; 18 to 23 months: aOR = 0.16, 95%CI: 0.05–0.48; 24 to 29 months: aOR = 0.26, 95%CI: 0.09–0.77; 30 to 35 months: aOR = 0.30, 95%CI: 0.11–0.85). Children living in urban areas with one sibling were also found to have an increased odds of non-uptake compared with children with none (aOR = 1.86, 95%CI: 1.03–3.33). For children living in rural areas, RV1 non-uptake was highest in children from the richest families (aOR = 4.40, 95%CI: 1.41–13.70), while children living in Lori had almost nine times the odds of non-uptake compared with those living in Armavir (aOR = 8.74, 95%CI: 1.47–52.13).

For RV2, among the children living in urban areas, older age was again associated with lower odds of non-uptake (6 to 11 months: aOR = 0.30, 95%CI: 0.12–0.75; 12 to 17 months: aOR = 0.17, 95%CI: 0.06–0.53; 18 to 23 months: aOR = 0.14, 95%CI: 0.04–0.47; 24 to 29 months: aOR = 0.22, 95%CI: 0.06–0.78; 30 to 35 months: aOR = 0.24, 95%CI: 0.08–0.72). Additionally, among the children living in urban areas, those whose mothers viewed distance to the health clinic as a big problem had almost three times the odds of non-uptake as those who did not (aOR = 2.94, 95%CI: 1.09–7.89). Among the children living in rural areas, non-uptake was highest in wealthier children, with those in the high wealth index category having almost four times the odds of non-uptake (aOR = 3.76, 95%CI: 1.56–9.05).

Table 4 presents the final multilevel models for each PCV dose. For PCV1, non-uptake was highest in the youngest children, those with one sibling, and those whose parents viewed distance to the health clinic as a problem. Compared with those aged 0 to 5 months, lower odds of non-uptake were seen among children aged 6 to 11 months (aOR = 0.20, 95%CI: 0.07–0.61) and children aged 12 to 17 months (aOR = 0.07, 95%CI: 0.005–0.88). Children with one sibling had almost three times the odds of not receiving PCV1 compared with those with no siblings (aOR = 2.96, 95%CI: 1.21–7.24) and children whose mothers defined the distance to the health clinic as a big problem had almost 11 times the odds of non-uptake (aOR = 10.93, 95%CI: 1.11–107.68). Children living in Aragatsotn were more likely to have received PCV1 compared with those living in Armavir (aOR = 0.02, 95%CI: <0.001–0.58). The non-uptake of PCV2 was highest in those with one sibling and in those whose parents viewed distance to the health clinic as a problem. Children with one sibling had over three times the odds of non-uptake compared to those with none (aOR = 3.18, 95%CI: 1.33–7.58) and those who reported distance to the health clinic as a big problem had about nine times the odds of non-uptake (aOR = 9.10, 95%CI: 2.30–26.10). Compared with Armavir, children living in Aragatsotn (aOR = 0.02, 95%CI: <0.001–0.44), Kotayk (aOR = 0.14, 95%CI: 0.02–0.94), and Shirak (aOR = 0.12, 95%CI: 0.02–0.70) had lower odds of not receiving PCV2. Only age was found to be associated with the non-uptake of PCV3, again with non-uptake being lower in children aged 6 to 11 months (aOR = 0.08, 95%CI: 0.03–0.22) and children aged 12 to 17 months (aOR = 0.03, 95%CI: 0.01–0.18) compared with children aged 0 to 5 months. 

Significant between-cluster variability was found for both RV doses among the children living in urban areas (Table 5). For RV1, 30.5% of the total variance in the odds of not receiving the vaccination was due to the variability between clusters (τ = 1.4416, *p* = 0.03), while 22.8% of the total variance in the odds of not receiving RV2 was due to the variability between clusters (τ = 1.3324, *p* = 0.02). For the second PCV dose, 53.9% of the total variance in the odds of not receiving the vaccination was due to variability between clusters (τ = 3.8479, *p* = 0.03). All other null ICC estimates were not significant.

## 4. Discussion

The 2015–2016 ADHS is the first data collection cycle after the RV and the PCV were added into Armenia’s National Immunization Schedule providing an opportunity to examine the initial uptake of both vaccines. The uptake of the RV was found to be high, but the uptake of the PCV was much lower. The initial uptake of the RV was in line with coverage levels of the other childhood vaccinations (tuberculosis (BCG), diphtheria–tetanus–pertussis (DTP), polio (Pol), and measles containing vaccine (MCV)), which ranged from 76.1% for the third dose of DTP to 91.7% for BCG in the 2015–2016 ADHS. PCV uptake, although lower, followed the same pattern as the other vaccines which require a course of three vaccinations (DTP and Pol). This may suggest that the two NVIs examined in this study are on track to have relatively high coverage levels once they become more established components of the National Immunization Schedule. Although more time has passed since the introduction of the RV compared to the PCV, which may have affected uptake, other factors (such as those examined in this study) may also be involved. Administering the RV orally as opposed to intramuscularly may be a reason for the differences in uptake, as this mode of delivery can be more appealing, especially in light of the multiple vaccinations children receive at the recommended ages. The high initial uptake of both RV doses is encouraging for the prospects of increasing PCV uptake. PCV1 and PCV2 are due at the same times as RV1 and RV2, demonstrating that mothers are interacting with the healthcare system around the recommended time. In an effort to increase the uptake of the PCV, healthcare providers can utilize this interaction with mothers and stress the importance of receiving all vaccinations as recommended in an attempt to overcome any potential apprehension among mothers during their visits.

When examining the factors associated with the non-uptake of the RV, a significant interaction between residence and wealth index was found, suggesting an inequity based on wealth index in accessing immunization resources among children living in rural regions in Armenia. Previous research on this topic is mixed, with differences possibly explained by donor assistance in low- and middle-income countries that support immunization campaigns targeted towards children from poorer households [13,18]. Due to support from Gavi during the initial implementation of NVIs, children living in the low wealth index category may have been targeted by campaigns, leaving those in the high wealth index category without the same resources, thus resulting in an observed disparity in uptake. Immunization services should ensure that resources are equitably available regardless of economic background to overcome such potential issues.

A number of other factors were found to be significantly associated with the non-uptake of both NVIs, in line with previous studies examining the uptake of vaccinations [12,13,16,17,18,19,20]. For several of the vaccinations, non-uptake was highest in the youngest children, implying that children are receiving their vaccinations later than recommended. Future studies should examine the timeliness of these vaccinations, specifically among urban children, to gain a better understanding of the dynamics behind these differences. Similar to previous studies [12,16,18,19,20], siblingship was also found to be a significant factor for some vaccinations, which may be reflective of the competing priorities that mothers with multiple children face. Additionally, the results indicate that the distance to a health facility is associated with the non-uptake of both the RV and the PCV. Distance to a health facility may be seen as a barrier to accessing healthcare services since it requires more time and resources than someone may have available. Potential strategies to overcome these issues that ought to be considered are mobile vaccination clinics or vaccination services in schools. These could address the barriers of distance and provide a much-needed service to these women and children.

The non-uptake of both NVIs also varied throughout the regions. This finding was in line with multiple studies that have found residence to be a factor in vaccination uptake and coverage [11,12,13,14,15]. Stratified analyses for the RV resulted in small sample sizes which may have led to imprecise confidence intervals for some estimates. Different methods of combining regions in an effort to strengthen the ability of these analyses in order to detect regional variations in uptake were examined, yet none were found to be sensible. Taking this into consideration, regions were kept in the models as there may be province-level characteristics that impact uptake. Further studies on how these NVIs were implemented in each region may provide more insight as to why these differences were seen.

Multilevel logistic regression models also allowed us to examine the effect of clustered sampling on the uptake of NVIs by calculating the ICC. For both doses of the RV, there was evidence of significant variability in the odds of non-uptake due to intra-cluster differences among children living in urban areas. For the PCV series, only PCV2 demonstrated significant variability. The observed decreases in the ICCs for RV1 and RV2 in rural areas and for PCV2 suggest that the included factors explain some of the variability in the non-uptake of these vaccinations between clusters. For both doses of the RV, among children living in urban areas the ICC did not decrease after adding explanatory variables. The inclusion of individual- and community-level factors should have had an impact on the ICCs since they are expected to explain some of the variance in the outcome of the model. The lack of an observed change indicates that factors other than those which were included in this analysis may be responsible for intra-cluster variability. Further studies on the differences between clusters could provide helpful information for future NVIs.

There are some limitations to this analysis that are important to consider. Vaccination data were based solely on the information recorded by trained field workers from cards completed by healthcare professionals, and the accuracy of this transcription is unknown. A vaccine that is not documented may be delayed but eventually received. Additionally, the sample was restricted to only those with documented vaccination information in order to curb recall bias, but this does usually introduce potential bias if having a vaccination card is associated with being vaccinated. Since most children presented health cards to the survey staff, this may not have been the case. There were some vaccinations reported by mothers that were not considered to be received, as there was no documented date, in an effort to have a conservative estimate of the uptake. The frequency of a child having only a maternal report of receipt of a vaccination was low and thus should not have a major impact on the estimates. Therefore, by restricting analyses to children with health cards and considering only those with a documented date of vaccination as received, uptake may be underestimated. Additionally, the 2015–2016 ADHS only allows for a relatively short timeframe when evaluating NVI implementation, and more time between introduction and analysis may be needed to provide a better understanding of uptake. Nonetheless, this study provides a baseline for future studies centered around the uptake of the RV and the PCV and sheds light on the mechanisms behind NVIs. Potential factors were limited to what was collected in the ADHS, and, consequently, other factors not evaluated may be associated with the uptake of the RV and the PCV. Furthermore, due to the small sample sizes in some regions, which limited the ability to evaluate the association of regions with non-uptake, associations observed should be taken as an indication for further studies with larger sample sizes in each region in order to obtain more conclusive results. Lastly, due to the cross-sectional nature of the data, causal inferences cannot be made, and factors that are found to be significant should thus be classified in association with the non-uptake of the RV and the PCV.

## 5. Conclusions

Using the most recent ADHS data, we were able to examine the uptake and associated factors of the two most recent NVIs among children living in Armenia. Although uptake was high for the RV, there is room for improvement for the PCV series, especially the third dose. Future studies should examine the impact of these vaccinations on the occurrence of related diseases in Armenia and assess the timing of each dose of both vaccines in order to provide more information on the uptake and coverage of these NVIs. A deeper look into other potential factors, such as structural factors, as well as maternal attitudes and knowledge, may offer insights that could be used to strengthen NVI activities among children living in Armenia. A better understanding of the uptake of the RV and the PCV is crucial in order to promote optimal coverage when using these vaccinations and better protect the children, as well as the general population, in Armenia.

## Figures and Tables

**Table 1 vaccines-11-01719-t001:** Demographic characteristics of all eligible living children aged 0 to 35 months based on select vaccinations (Armenia, 2015–2016).

	RV (N = 1017)		PCV (N = 371)	
	n	Weighted % (95%CI)	n	Weighted % (95%CI)
**Sex**				
Male	534	51.1 (48.0–54.2)	197	50.2 (44.6–55.8)
Female	483	48.9 (45.8–52.0)	174	49.8 (44.24–55.4)
**Age (in months)**				
0–5	163	15.9 (13.5–18.2)	163	43.8 (38.4–49.2)
6–11	179	17.8 (15.3–20.3)	177	48.6 (43.0–54.2)
12–17	183	16.7 (14.5–19.0)	31	7.6 (4.5–10.7)
18–23	162	16.6 (13.8–19.5)	-	
24–29	190	19.0 (15.9–22.2)	-	
30–35	140	13.9 (11.0–16.8)	-	
**Siblingship**				
None	360	35.6 (32.2–39.0)	155	42.4 (36.4–48.3)
1	440	42.8 (39.1–46.6)	147	38.3 (32.4–44.2)
2+	217	21.6 (18.6–24.5)	69	2.3 (14.7–24.0)
**Birth Order**				
1	432	42.5 (39.4–45.6)	158	42.9 (37.0–48.8)
2	380	37.2 (34.0–40.4)	145	38.0 (32.2–43.9)
3+	205	20.3 (17.6–23.0)	68	19.0 (14.5–23.6)
**Age of mother (in years)** (mean, (sd); 95%CI for mean)	27.6 (0.2)	27.3–28.0	27.0 (0.3)	26.4–27.7
**Educational level of mother**				
Basic	65	6.7 (4.0–9.4)	18	5.3 (2.4–8.1)
Secondary	425	38.8 (35.0–42.6)	151	37.7 (32.5–43.0)
Higher	527	54.5 (50.6–58.4)	202	57.0 (51.8–62.1)
**Employment status of mother**				
Unemployed	824	80.7 (77.6–83.7)	298	79.6 (74.7–84.4)
Employed	193	19.3 (16.3–22.3)	73	20.4 (15.6–25.3)
**Antenatal care** ^a^				
Less than 4 visits or don’t know	32	3.4 (2.0–4.8)	11	2.8 (1.2–4.4)
At least 4 visits	892	96.6 (95.2–98.0)	353	97.2 (95.6–98.8)
**Place of delivery** ^b^				
Public hospital/maternity center	971	95.7 (94.1–97.3)	352	95.3 (92.0–98.5)
Private hospital/maternity center	39	4.3 (2.6–5.9)	14	4.7 (1.5–8.0)
**Wealth index** ^c^				
Poorest	206	19.4 (16.1–22.6)	60	15.3 (11.3–19.2)
Poorer	215	19.7 (16.8–22.7)	89	22.4 (17.9–27.0)
Middle	215	18.3 (15.1–21.5)	71	16.3 (12.3–20.3)
Richer	189	17.7 (14.2–21.2)	77	20.5 (15.2–25.9)
Richest	192	24.9 (19.0–30.8)	74	25.4 (18.1–32.7)
**Distance to health clinic**				
Not a big problem	899	89.7 (87.3–92.0)	333	91.0 (87.7–94.3)
Big problem	118	10.3 (8.0–12.7)	38	9.0 (5.7–12.3)
**Place of residence**				
Urban	576	58.0 (54.1–61.9)	219	61.2 (57.1–65.2)
Rural	441	41.9 (38.0–45.8)	152	38.8 (34.8–42.9)
**Region**				
Yerevan	160	30.5 (25.9–35.1)	57	29.7 (24.0–35.4)
Aragatsotn	44	3.3 (2.4–4.3)	19	4.5 (3.4–5.6)
Ararat	119	10.4 (8.2–12.6)	32	6.7 (5.1–8.2)
Armavir	124	11.7 (9.5–13.9)	49	12.2 (10.0–14.5)
Gegharkunik	50	4.2 (3.2–5.2)	21	4.7 (3.7–5.8)
Lori	50	5.2 (3.9–6.6)	20	5.9 (4.7–7.1)
Kotayk	136	13.6 (10.9–16.3)	54	15.1 (12.2–17.9)
Shirak	100	9.6 (7.8–11.3)	38	9.9 (7.9–12.0)
Syunik	59	3.5 (2.8–4.3)	25	4.0 (3.0–5.0)
Vayots Dzor	69	2.0 (1.5–2.5)	18	1.4 (1.0–1.8)
Tavush	106	5.9 (4.5–7.3)	38	5.7 (4.5–7.0)

CI: confidence interval; RV: rotavirus vaccine series; PCV: pneumococcal conjugate vaccine series. ^a^ Antenatal care missing in 93 for the RV and 7 for the PCV. ^b^ Place of delivery missing in 7 for the RV and 5 for the PCV. ^c^ Composite measures generated by principal component analysis based on the household’s ownership of certain assets, housing construction materials, and access to water and sanitation.

**Table 2 vaccines-11-01719-t002:** Vaccination information among all living children aged 0 to 35 months eligible for select vaccinations (Armenia, 2015–2016).

	Received n Weighted % (95%CI)	Method Reported n Weighted % (95%CI)			Not Received n Weighted % (95%CI)
		Date Recorded on Card	Reported by Mother	Marked on Card (No Date)	
**RV1** ^a^ (N = 1017)	905 90.0 (87.9–92.1)	860 84.4 (81.5–87.3)	44 4.0 (2.3–5.7)	1 0.1 (0–0.4)	97 9.8 (7.8–11.8)
**RV2** ^b^ (N = 958)	821 86.6 (83.8–89.5)	784 82.0 (78.8–85.2)	35 2.9 (1.7–4.1)	2 0.2 (0–0.5)	123 13.1 (10.4–15.9)
**PCV1** ^c^ (N = 371)	307 83.5 (79.0–88.0)	293 78.7 (73.8–83.5)	13 3.2 (1.3–5.2)	1 0.4 (0–1.1)	59 16.2 (11.8–20.7)
**PCV2** ^d^ (N = 313)	248 79.4 (73.7–85.2)	238 75.2 (69.2–81.2)	8 2.2 (0.7–3.8)	2 0.7 (0–1.7)	60 20.2 (14.5–25.8)
**PCV3** ^e^ (N = 285)	162 57.5 (50.9–64.2)	160 55.5 (48.9–62.1)	1 0.5 (0–1.4)	1 0.5 (0–1.5)	118 41.6 (35.0–48.3)

RV1: rotavirus vaccine, dose 1; RV2: rotavirus vaccine, dose 2; PCV1: pneumococcal conjugate vaccine, dose 1; PCV2: pneumococcal conjugate vaccine, dose 2; PCV3: pneumococcal conjugate vaccine, dose 3. ^a^ RV1 missing in 15 children. ^b^ RV2 missing in 16 children. ^c^ PCV1 missing in 5 children. ^d^ PCV2 missing in 5 children. ^e^ PCV3 missing in 5 children.

**Table 3 vaccines-11-01719-t003:** Factors associated with the non-uptake of the rotavirus vaccine series among eligible children aged 0 to 35 months (Armenia, 2015–2016).

	RV1 (N = 1017)		RV2 (N = 958)	
	Urban aOR (95%CI)	Rural aOR (95%CI)	Urban aOR (95%CI)	Rural aOR (95%CI)
**Sex**				
Male	Ref			
Female	0.59 (0.33–1.04)			
**Age (in months)**				
0–5	Ref	Ref	Ref	
6–11	0.34 (0.14–0.84) *	0.71 (0.23–2.22)	0.30 (0.12–0.75) *	
12–17	0.18 (0.04–0.77) *	0.38 (0.11–1.33)	0.17 (0.06–0.53) *	
18–23	0.16 (0.05–0.48) *	0.30 (0.07–1.41)	0.14 (0.04–0.47) *	
24–29	0.26 (0.09–0.77) *	0.19 (0.07–0.52) *	0.22 (0.06–0.78) *	
30–35	0.30 (0.11–0.85) *	0.50 (0.15–1.69)	0.24 (0.08–0.72) *	
**Siblingship**				
None	Ref			
1	1.86 (1.03–3.33) *			
2+	0.83 (0.37–1.90)			
**Wealth Index**				
Low		Ref		Ref
Middle		1.40 (0.55–3.56)		2.07 (0.78–5.48)
High		4.40 (1.41–13.70) *		3.76 (1.56–9.05) *
**Distance to health clinic**				
Not a big problem			Ref	
Big problem			2.94 (1.09–7.89) *	
**Region**				
Yerevan		-	0.94 (0.36–2.45)	-
Aragatsotn		<0.001 (<0.001–<0.001) **	1.11 (0.18–6.96)	<0.001 (<0.001–<0.001) **
Ararat		0.85 (0.24–3.04)	0.68 (0.21–2.19)	1.20 (0.49–2.98)
Armavir		Ref	Ref	Ref
Gegharkunik		0.25 (0.03–2.00)	2.58 (0.72–9.20)	0.56 (0.12–2.70)
Lori		8.74 (1.47–52.13) *	1.26 (0.30–5.38)	4.40 (0.90–21.58)
Kotayk		0.55 (0.12–2.63)	0.11 (0.02–5.38)	0.50 (0.09–2.66)
Shirak		0.36 (0.08–1.57)	0.34 (0.07–1.70)	0.35 (0.10–1.20)
Syunik		0.74 (0.15–3.67)	1.28 (0.48–3.40)	0.56 (0.06–4.93)
Vayots Dzor		0.88 (0.21–3.77)	0.69 (0.16–2.92)	1.70 (0.60–4.81)
Tavush		1.12 (0.36–3.49)	1.70 (0.56– 5.20)	0.77 (0.28–2.12)

RV1: rotavirus vaccine, first dose; RV2: rotavirus vaccine, second dose; aOR: adjusted odds ratio; CI: confidence interval. * *p* < 0.05; ** *p* < 0.0001.

**Table 4 vaccines-11-01719-t004:** Factors associated with the non-uptake of the pneumococcal conjugate vaccine series among eligible living children aged 0 to 35 months (Armenia, 2015–2016).

	PCV1 (N = 371) aOR (95%CI)	PCV2 (N = 313) aOR (95%CI)	PCV3 (N = 285) aOR (95%CI)
**Gender**			
Male		Ref	
Female		2.07 (0.95–4.51)	
**Age (in months)**			
0–5	Ref		Ref
6–11	0.20 (0.07–0.61) *		0.08 (0.03–0.22) **
12–17	0.07 (0.005–0.88) *		0.03 (0.01–0.18) *
**Siblingship**			
None	Ref	Ref	
1	2.96 (1.21–7.24) *	3.18 (1.33–7.58) *	
2+	0.65 (0.18–2.37)	1.15 (0.34–3.88)	
**Educational level of mother**			
Basic	Ref		
Secondary	0.14 (0.02–1.03)		
Higher	0.14 (0.02 -1.03)		
**Wealth Index**			
Low		Ref	
Middle		0.79 (0.26–2.41)	
High		1.81 (0.63–5.24)	
**Distance to health clinic**			
Not a big problem	Ref	Ref	
Big problem	10.93 (1.11–107.68) *	9.10 (2.30–36.10) *	
**Region**			
Yerevan	0.71 (0.11–4.70)	0.89 (0.19–4.09)	
Aragatsotn	0.02 (<0.001–0.58) *	0.02 (<0.001–0.44) *	
Ararat	0.08 (0.01–1.25)	0.23 (0.04–1.26)	
Armavir	Ref	Ref	
Gegharkunik	0.90 (0.07–11.83)	1.76 (0.27–11.34)	
Lori	4.20 (0.49–35.77)	2.32 (0.40–13.42)	
Kotayk	0.21 (0.02–1.73)	0.14 (0.02–0.94) *	
Shirak	0.26 (0.03–2.51)	0.12 (0.02–0.70) *	
Syunik	0.80 (0.08–8.15)	2.79 (0.60–13.08)	
Vayots Dzor	0.37 (0.03–4.80)	1.11 (0.19–6.64)	
Tavush	0.36 (0.03–4.37)	0.29 (0.04–1.96)	

PCV1: pneumococcal conjugate vaccine, first dose; PCV2: pneumococcal conjugate vaccine, second dose; PCV3: pneumococcal conjugate vaccine, third dose; aOR: adjusted odds ratio; CI: confidence interval. * *p* < 0.05; ** *p* < 0.0001.

**Table 5 vaccines-11-01719-t005:** Intra-cluster correlations (ICCs) for the multilevel logistic regression model based on the uptake of the rotavirus vaccine (RV) series and the pneumococcal conjugate vaccine (PCV) series among eligible living children aged 0 to 35 months (Armenia, 2015–2016).

	Null Model	Model with Individual-Level Factors Only	Model with Community-Level Factors Only	Final Multivariable Model
**RV1—Urban**	0.3047 *	0.3552 *	0.2616 *	0.3588 *
**RV1—Rural**	0.3161	0.4880	0.1470	0.1870
**RV2—Urban**	0.2282 *	0.2941 *	0.2272 *	0.2557 *
**RV2—Rural**	0.2099	0.3117	0.0544	0.0574
**PCV1**	0.3475	0.5029	0.2119	0.5082
**PCV2**	0.5391 *	0.4389	0.3664	0.2352
**PCV3**	0.3371	0.3849	0.1902	0.5106 *

PCV1: pneumococcal conjugate vaccine, first dose; PCV2: pneumococcal conjugate vaccine, second dose; PCV3: pneumococcal conjugate vaccine, third dose. * *p* < 0.05. ICC = τ_00_/[τ_00_ + (π^2^/3)], where τ_00_ is the covariance parameter estimate generated from the model.

## Data Availability

Data are contained within the article.

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
