# Peer review of "Factors Associated with the Uptake of Rotavirus and Pneumococcal Conjugate Vaccines among Children in Armenia: Implications for Future New Vaccine Introductions"

_vaccines, 2023, doi:10.3390/vaccines11111719_

Round 1
Reviewer 1 Report (New Reviewer)
Comments and Suggestions for Authors
Overview: In this article, the authors analyze the uptake of rotavirus and pneumococcal conjugate vaccines following their introductions in Armenia, using data from the 2015-2016 Armenia DHS. Overall, the manuscript is well-written, organized, and easy to follow. The topic is of interest both for policymakers in Armenia and in others interested in the factors that may be associated with uptake of relatively recently-introduced vaccines. I have a few methodological questions and several more minor comments, listed below:
Major comments:
1. Lines 83-84 state that all living children aged 0-35 months with documented birthdates were used as the analytic sample for this analysis. Did the authors exclude those too young for vaccination at the time of the survey, e.g. < 6 weeks for the first dose, etc.? If not, why not? It seems like those children wouldn’t contribute any useful information to the analysis, as they are not yet eligible for vaccination.
This ties into the discussion in lines 248-252, which comments on the analyses that are using 0-5 months old as a reference group and calls for efforts to enhance timeliness. This may certainly be true, but in order to address this question, children younger than the age of eligibility would need to be excluded from the analysis. For instance, as PCV3 isn’t recommended until age 18 weeks (lines 60-61), even with perfect adherence to vaccination schedules, the coverage in 0-5 month-olds will be low (as children in the first ~4 months of life aren’t yet eligible).
2. Table 3: I am somewhat confused by the models that are presented here – I see that the urban RV1 model, for instance, includes sex and siblingship, whereas none of the other models do; the rural models includes wealth index but the urban models do not, the urban RV2 model is the only one that includes distance to health clinic. From the methods section and these results, I am guessing that the univariable models used for covariate selection as described in lines 113-116 were fit for each different outcome of interest, and hence the authors ended up with different combinations of covariates for the various multilevel multivariable models – is that correct?
If so, it does make it difficult to directly compare the relationships found in the tables, since each of the aORs for the different models is adjusted for a different set of factors. The authors already have done a nice job of assessing for collinear predictors using their VIF analysis and developing a principled set of possible predictors by reviewing the literature. Variable selection can be helpful to deal with collinearity, but the VIF process should be a good step towards removing this risk. I am not sure that the covariate selection step is strictly necessary, therefore.
The authors’ goal seems to be primarily one of inference, rather than predictive performance – which is appropriate given the questions that they are seeking to address. Univariable regression-based variable selection processes like the ones that the authors use here, however, may complicate inference. In addition to the challenges in comparability between models as described above, there is the possibility that factors that are below a given threshold of significance in univariable analysis (I think that the authors are using p = 0.05 here) may actually be potentially significant in a multivariable analysis, e.g. when the relationship between the outcome and the covariate is confounded by one of the other covariates in the analysis. See e.g. Sun, Shook, and Kay (J Clin Epidemiol 1996 49;8:907-916), among others.
It could be argued, therefore, that it would be more appropriate to forgo the covariate selection step and instead fit models that include all of the selected predictors (while accounting for collinearity via an approach like the VIF analysis that the authors have already done). Did the authors consider using the same set of predictors for each of their analyses, instead of the univariable regression variable selection approach? I would be interested to hear the authors’ thoughts on the decision to include this variable selection step and implications for inference.
Minor comments:
1. Lines 40-41: “Unfortunately, NVIs have also been observed to have a negative impact 40 on health systems as a result of inadequate planning and, consequently, logistical issues 41 and vaccine shortages.” I would suggest that the authors add a reference for this statement if possible.
2. Lines 70-73: perhaps this seems obvious from the introduction, but I would suggest that the authors specify that they are using the 2015-2016 Armenia DHS when they first introduce the ADHS (e.g. in line 73 could just write “The 2015-2016 ADHS”). This would help the reader clearly know which of the ADHS rounds is being used here.
3. Lines 113-120: Can the authors clarify how “community” was defined and used in these analyses?
3a. It would be helpful perhaps in lines 100-103 to specify which of these were treated as community vs individual-level covariates. It seems like place of residence and region may have been the only community-level predictors
3b. How were the “individual- and community-level models” in lines 115-116 constructed? Did the authors somehow aggregate the coverage to the community level before testing the community level predictors? Or does this just indicate that both the individual and community predictors were included in models of individual-level vaccination coverage?
3c. What was used to define “community” for the community-level random effects in lines 117-119 , which are then used to assess ICC? Were these the survey PSUs, cities / villages based on place names, some administrative division (districts), etc.? It would be helpful to specify how this was defined explicitly in the methods to help with the interpretation of the ICCs and related findings in the results.
4. Table 3: Distance to health clinic is given as “not a big problem” vs “big problem”. I would have expected this to be a measurement of distance – but perhaps this was actually asked of each surveyed individual as a qualitative assessment? I might suggest that the authors provide a little more information about this variable in the methods section or in a caption for the figure.
5. Lines 247-259: I would suggest that the authors refer to other similar studies that have investigated some of these risk factors for poor vaccine uptake, e.g. distance to health facilities, siblingship, etc. – are the findings in Armenia similar to those seen in other settings?
6. Line 285: Here, the authors state that they excluded individuals without vaccination cards. This seems to conflict with the discussion in the methods (lines 88-94), which says that “if neither was available, vaccination history reported by the mother were [should be “was”?] recorded”, and Table 2, which includes the % reported by mother. For RV1, for instance, adding the # received (905, including 44 reported by mother), not received (97), and missing (15) from Table 2 produces the 1,017 children included in Table 1 and used in the denominator for table 2. If the authors were indeed limiting the analysis to only those individuals with cards, then it would be good to make this more clear in the methods and be more clear in the results which were conducted using recall+card and which were restricted to card-only.
7. General comment: I appreciate that no survey data are available since the 2015-2016 DHS in Armenia, and so the authors are constrained with respect to the recency of the analysis. Given that these data are now 7 years old, I wonder if the authors might consider commenting in the discussion on any trends that have been observed since the DHS, e.g. in official country reported data or administrative data (even though those data also have limitations)?
Author Response
Overview: In this article, the authors analyze the uptake of rotavirus and pneumococcal conjugate vaccines following their introductions in Armenia, using data from the 2015-2016 Armenia DHS. Overall, the manuscript is well-written, organized, and easy to follow. The topic is of interest both for policymakers in Armenia and in others interested in the factors that may be associated with uptake of relatively recently-introduced vaccines. I have a few methodological questions and several more minor comments, listed below:
Major comments:
- Lines 83-84 state that all living children aged 0-35 months with documented birthdates were used as the analytic sample for this analysis. Did the authors exclude those too young for vaccination at the time of the survey, e.g. < 6 weeks for the first dose, etc.? If not, why not? It seems like those children wouldn’t contribute any useful information to the analysis, as they are not yet eligible for vaccination.
This ties into the discussion in lines 248-252, which comments on the analyses that are using 0-5 months old as a reference group and calls for efforts to enhance timeliness. This may certainly be true, but in order to address this question, children younger than the age of eligibility would need to be excluded from the analysis. For instance, as PCV3 isn’t recommended until age 18 weeks (lines 60-61), even with perfect adherence to vaccination schedules, the coverage in 0-5 month-olds will be low (as children in the first ~4 months of life aren’t yet eligible).
All those too young for each vaccination at the time of the survey were excluded from the analysis for that particular vaccination. As stated in lines 84-87 “The subsamples for each vaccination were composed of children eligible for that particular vaccination and dose based on the recommendations of Armenia’s National Immunization Program schedule and year the vaccine was introduced.” This exclusion was applied per vaccination to all analyses such that the examination for each vaccination (RV1, RV2, PCV1, PCV2, or PCV3) was based on an analytic sample that only included children eligible for the vaccination. By using the youngest as the reference, we also were able to get a sense of timeliness of these vaccines to understand if those who were older had lower uptake and coverage and thus may need more focus in vaccination campaigns.
- Table 3: I am somewhat confused by the models that are presented here – I see that the urban RV1 model, for instance, includes sex and siblingship, whereas none of the other models do; the rural models includes wealth index but the urban models do not, the urban RV2 model is the only one that includes distance to health clinic. From the methods section and these results, I am guessing that the univariable models used for covariate selection as described in lines 113-116 were fit for each different outcome of interest, and hence the authors ended up with different combinations of covariates for the various multilevel multivariable models – is that correct?
If so, it does make it difficult to directly compare the relationships found in the tables, since each of the aORs for the different models is adjusted for a different set of factors. The authors already have done a nice job of assessing for collinear predictors using their VIF analysis and developing a principled set of possible predictors by reviewing the literature. Variable selection can be helpful to deal with collinearity, but the VIF process should be a good step towards removing this risk. I am not sure that the covariate selection step is strictly necessary, therefore.
The authors’ goal seems to be primarily one of inference, rather than predictive performance – which is appropriate given the questions that they are seeking to address. Univariable regression-based variable selection processes like the ones that the authors use here, however, may complicate inference. In addition to the challenges in comparability between models as described above, there is the possibility that factors that are below a given threshold of significance in univariable analysis (I think that the authors are using p = 0.05 here) may actually be potentially significant in a multivariable analysis, e.g. when the relationship between the outcome and the covariate is confounded by one of the other covariates in the analysis. See e.g. Sun, Shook, and Kay (J Clin Epidemiol 1996 49;8:907-916), among others.
It could be argued, therefore, that it would be more appropriate to forgo the covariate selection step and instead fit models that include all of the selected predictors (while accounting for collinearity via an approach like the VIF analysis that the authors have already done). Did the authors consider using the same set of predictors for each of their analyses, instead of the univariable regression variable selection approach? I would be interested to hear the authors’ thoughts on the decision to include this variable selection step and implications for inference.
This is correct. We fit univariable models for each outcome of interest (uptake of RV1, RV2, PCV1, PCV2, or PCV3). If there was not an estimate for a particular covariate listed on the table, then it was not included on the model based on the univariable analyses. We did consider including the same set of covariates for all models, but we chose to proceed with model selection in this way in order to not overfit the model and include the most appropriate potential covariates. We had also included all covariates in order to compare the results from the model we had selected and that of one including all covariates and saw that they did not change the final results, and thus in order to have the most parsimonious model we chose to proceed with the initial univariable analysis to form our models. As for comparison between models, our goal was to be able to infer if any potential covariates affected uptake of each specific vaccination, in order to inform policy and practice. We were not specifically interested in comparing covariates that affect uptake across vaccinations, and we believe that examining different covariates in each model was appropriate for this approach.
Minor comments:
- Lines 40-41: “Unfortunately, NVIs have also been observed to have a negative impact 40 on health systems as a result of inadequate planning and, consequently, logistical issues 41 and vaccine shortages.” I would suggest that the authors add a reference for this statement if possible.
Included. See line 42.
- Lines 70-73: perhaps this seems obvious from the introduction, but I would suggest that the authors specify that they are using the 2015-2016 Armenia DHS when they first introduce the ADHS (e.g. in line 73 could just write “The 2015-2016 ADHS”). This would help the reader clearly know which of the ADHS rounds is being used here.
Included. See line 73.
- Lines 113-120: Can the authors clarify how “community” was defined and used in these analyses?
Community-level factors were factors that were present on a more societal level, rather than individual. Thus, any factor that was collected by the DHS that could affect uptake that was specific to the child or mother was considered “individual”, and any factor collected by the DHS that could affect uptake that was related to their environment was considered “community”.
3a. It would be helpful perhaps in lines 100-103 to specify which of these were treated as community vs individual-level covariates. It seems like place of residence and region may have been the only community-level predictors.
Clarified. See line 100-107.
3b. How were the “individual- and community-level models” in lines 115-116 constructed? Did the authors somehow aggregate the coverage to the community level before testing the community level predictors? Or does this just indicate that both the individual and community predictors were included in models of individual-level vaccination coverage?
Separate models were fit for examining individual and community level factors’ association with vaccination uptake of each of the vaccinations (RV1, RV2, PCV1, PCV2, and PCV3). The reasoning for this was to determine how these community level factors impacted individual-level vaccination uptake.
3c. What was used to define “community” for the community-level random effects in lines 117-119 , which are then used to assess ICC? Were these the survey PSUs, cities / villages based on place names, some administrative division (districts), etc.? It would be helpful to specify how this was defined explicitly in the methods to help with the interpretation of the ICCs and related findings in the results.
We examined community-level effects at the PSUs/cluster level. Included in line 121.
- Table 3: Distance to health clinic is given as “not a big problem” vs “big problem”. I would have expected this to be a measurement of distance – but perhaps this was actually asked of each surveyed individual as a qualitative assessment? I might suggest that the authors provide a little more information about this variable in the methods section or in a caption for the figure.
This was how it was asked/documented in the DHS data. Clarified in lines 105-106.
- Lines 247-259: I would suggest that the authors refer to other similar studies that have investigated some of these risk factors for poor vaccine uptake, e.g. distance to health facilities, siblingship, etc. – are the findings in Armenia similar to those seen in other settings?
Our findings were in line with previous studies examining factors associated with uptake of vaccinations. Included in lines 250-252.
- Line 285: Here, the authors state that they excluded individuals without vaccination cards. This seems to conflict with the discussion in the methods (lines 88-94), which says that “if neither was available, vaccination history reported by the mother were [should be “was”?] recorded”, and Table 2, which includes the % reported by mother. For RV1, for instance, adding the # received (905, including 44 reported by mother), not received (97), and missing (15) from Table 2 produces the 1,017 children included in Table 1 and used in the denominator for table 2. If the authors were indeed limiting the analysis to only those individuals with cards, then it would be good to make this more clear in the methods and be more clear in the results which were conducted using recall+card and which were restricted to card-only.
All analyses were restricted to children with vaccination cards, but there were children with incomplete vaccination cards which is why there are some that had documentation by maternal report. A vaccination was deemed as received only if there was documentation of receipt (not maternal report of receipt). The description in the methods was included to provide the reader full information on the methods used in the DHS to assess vaccination status. The tables were also included to provide the reader a better understanding of reported vaccinations in the DHS. All regression models were restricted to children eligible for each vaccination with documented vaccination status. Clarified in lines 100-102.
- General comment: I appreciate that no survey data are available since the 2015-2016 DHS in Armenia, and so the authors are constrained with respect to the recency of the analysis. Given that these data are now 7 years old, I wonder if the authors might consider commenting in the discussion on any trends that have been observed since the DHS, e.g. in official country reported data or administrative data (even though those data also have limitations)?
We agree that it would be helpful to include more recent data for the reader to better understand the findings. Unfortunately, there are no publicly available data on vaccination coverage. The authors are currently working with the local authorities in Armenia to obtain the most recent data in order to conduct follow-up studies.
Reviewer 2 Report (New Reviewer)
Comments and Suggestions for Authors
The authors have examined factors associated with RV and PCV uptake in Armenia, using the 2015-16 DHS dataset. It's a cross-sectional analysis with standard measures collected in the survey. I have a few questions and suggestions:
1) I am not certain why the authors have chosen the 0-5 months of age for the reference group for all the vaccines. Are these vaccines even indicated for this age group? The authors should exclude children who were not eligible for the vaccine and re-think their referent age group.
2) I'm confused by the PCV3 and RV2 Rural analyses in tables 3-4. They are just blank for a number of parameters. Are these parameters not included in the models? Was this an error?
3) The authors need to include sample sizes in the tables.
4) I'd like to see a comparison of RV and PCV vaccines to other childhood vaccines (measles, polio, dpt, etc)
Author Response
The authors have examined factors associated with RV and PCV uptake in Armenia, using the 2015-16 DHS dataset. It's a cross-sectional analysis with standard measures collected in the survey. I have a few questions and suggestions:
1) I am not certain why the authors have chosen the 0-5 months of age for the reference group for all the vaccines. Are these vaccines even indicated for this age group? The authors should exclude children who were not eligible for the vaccine and re-think their referent age group.
All those too young for each vaccination at the time of the survey were excluded from the analysis. As stated in lines 84-87 “The subsamples for each vaccination were composed of children eligible for that particular vaccination and dose based on the recommendations of Armenia’s National Immunization Program schedule and year the vaccine was introduced.” This exclusion was applied per vaccination to all analyses such that the examination for each vaccination (RV1, RV2, PCV1, PCV2, or PCV3) was based on an analytic sample that only included children eligible for the vaccination. By using the youngest as the reference, we also were able get a sense of timeliness of these vaccines to understand if those who were older had lower coverage and thus may need more focus in vaccination campaigns.
2) I'm confused by the PCV3 and RV2 Rural analyses in tables 3-4. They are just blank for a number of parameters. Are these parameters not included in the models? Was this an error?
We fit univariable models for each outcome of interest (uptake of RV1, RV2, PCV1, PCV2, or PCV3). If there is not an estimate for a particular covariate listed on the table, then it was not included on the model based on the univariable analyses. We did consider including the same set of covariates for all models, but we chose to proceed with model selection in this way in order to not overfit the model and include the most appropriate potential covariates. We had also included all covariates in order to compare the results from the model we had selected and that of one including all covariates and saw that they did not change the final results, and thus in order to have the most parsimonious model we chose to proceed with the initial univariable analysis to form our models. As for comparison between models, our goal was to be able to infer if any potential covariates affected uptake of each specific vaccination, in order to inform policy and practice. We were not specifically interested in comparing covariates that affect uptake across vaccinations, and we believe that examining different covariates in each model was appropriate for this approach.
3) The authors need to include sample sizes in the tables.
Included. Please refer to the tables.
4) I'd like to see a comparison of RV and PCV vaccines to other childhood vaccines (measles, polio, dpt, etc)
We too believe this is a necessary analysis, but since there had been no studies examining the initial uptake we decided to first assess uptake. As the data were collected quite soon after introduction, we were able to capture initial uptake of the vaccination which can provide information for future new vaccine introductions. This also would make it a bit difficult to compare the uptake and coverage of RV and PCV to the more established recommended vaccinations. It is an analysis that perhaps now would warrant a study if we can find suitable data.
Reviewer 3 Report (New Reviewer)
Comments and Suggestions for Authors
It is a retrospective study regarding RV and PCV vaccine in a period of 1 year. Introduction is quite well written with all the theory parameters about these two vaccines. Materials and methods are well designed for the study. The study time of 1 year is short. The results are well analyzed with figures. Maybe the figures are so many with a lot of parameters. Some of them could be deleted.
In discussion, it is widely mentioned by the authors all the knowlegde about the current study. There are no enough literature references from similar studies that are mentioned or analyzed. Maybe some more need to be analyzed in discussion. The limitations that the authors are pointed out in discussion are really correct with a major importance for the study. In my opinion it is a good designed study for a country like Armenia but for a subject like PCV vaccine that have been studied a lot already in the literature.
Comments on the Quality of English LanguageMinor changes
Author Response
It is a retrospective study regarding RV and PCV vaccine in a period of 1 year. Introduction is quite well written with all the theory parameters about these two vaccines. Materials and methods are well designed for the study. The study time of 1 year is short. The results are well analyzed with figures. Maybe the figures are so many with a lot of parameters. Some of them could be deleted.
We do recognize that the time period of follow-up is relatively short, but we believe this provides an opportunity to examine the initial uptake of these new vaccine introductions and provide insights on factors associated with that initial uptake. We examined the uptake of 5 vaccinations overall (3 RV and 2 PCV) which resulted in quite a few analyses, and so we believe that each of the tables displays important information on the uptake of the individual vaccinations.
In discussion, it is widely mentioned by the authors all the knowlegde about the current study. There are no enough literature references from similar studies that are mentioned or analyzed. Maybe some more need to be analyzed in discussion. The limitations that the authors are pointed out in discussion are really correct with a major importance for the study. In my opinion it is a good designed study for a country like Armenia but for a subject like PCV vaccine that have been studied a lot already in the literature.
Our findings were in line with previous studies examining factors associated with uptake of vaccinations and references have been mentioned in the text. Included in line 244, 251-252, 256 and 265-267.
Round 2
Reviewer 1 Report (New Reviewer)
Comments and Suggestions for Authors
I thank the authors for their responses to my questions and comments, which have all been addressed. I have only a few small suggestions for clarification in language remaining.
Major comments:
Previous major comment #1: I appreciate the authors’ clarification that children too young for vaccination were excluded from the analysis, as is appropriate. As one of the other reviewers had the same question, I would suggest that the authors might consider adding this as a footnote to the relevant tables or changing from “0-35 months” to “eligible children < 36 months” in the table header to avoid confusion. No other questions in this area.
Previous major comment #2: I appreciate the authors’ response. As the authors are only interested in within-vaccination comparisons and predictors, this seems like a reasonable strategy. No further questions.
Minor comments:
The authors have thoroughly addressed my comments for minor comments 1-5 – no further questions.
Previous minor comment #6: I appreciate the clarification from the authors. In addition to the clarification provided in lines 100-102, I would suggest that the authors more explicitly state that all analyses were restricted to children with vaccination cards, and perhaps also provide some information about how often there were maternal reports of vaccine receipt that were discarded due to a lack of documentation (e.g. was this a relatively rare occurrence)?
Previous minor comment #7: Thanks to the authors for their response – I appreciate that there is are substantial gaps in recent vaccine coverage data in many countries and thank the authors for their ongoing work with authorities in Armenia to help rectify these gaps. No further questions on this topic.
Author Response
I thank the authors for their responses to my questions and comments, which have all been addressed. I have only a few small suggestions for clarification in language remaining.
Major comments:
Previous major comment #1: I appreciate the authors’ clarification that children too young for vaccination were excluded from the analysis, as is appropriate. As one of the other reviewers had the same question, I would suggest that the authors might consider adding this as a footnote to the relevant tables or changing from “0-35 months” to “eligible children < 36 months” in the table header to avoid confusion. No other questions in this area.
All table titles include a reference to “eligible living children 0 to 35 months”.
Previous major comment #2: I appreciate the authors’ response. As the authors are only interested in within-vaccination comparisons and predictors, this seems like a reasonable strategy. No further questions.
Noted, thank you.
Minor comments:
The authors have thoroughly addressed my comments for minor comments 1-5 – no further questions.
Noted, thank you.
Previous minor comment #6: I appreciate the clarification from the authors. In addition to the clarification provided in lines 100-102, I would suggest that the authors more explicitly state that all analyses were restricted to children with vaccination cards, and perhaps also provide some information about how often there were maternal reports of vaccine receipt that were discarded due to a lack of documentation (e.g. was this a relatively rare occurrence)?
Included. See line 101. Vaccination cards or “health cards” were seen in roughly 93% of children and as noted in Table 2, maternal report for each vaccination ranged from 0.5% to 4.4%, and thus these were considered not received for the analyses. Included. See line 301-307.
Previous minor comment #7: Thanks to the authors for their response – I appreciate that there is are substantial gaps in recent vaccine coverage data in many countries and thank the authors for their ongoing work with authorities in Armenia to help rectify these gaps. No further questions on this topic.
Noted, thank you.
Reviewer 2 Report (New Reviewer)
Comments and Suggestions for Authors
The authors could improve the manuscript by including the vaccination coverage from other childhood diseases, comparing uptake of RV and PCV to well established polio, measles, and DPT vaccines. The DHS data the authors are using has all these measures as well, so I don't understand the authors' resistance to including this. A logical step in examining uptake of a new vaccine is comparing it to other similar vaccines.
Author Response
The authors could improve the manuscript by including the vaccination coverage from other childhood diseases, comparing uptake of RV and PCV to well established polio, measles, and DPT vaccines. The DHS data the authors are using has all these measures as well, so I don't understand the authors' resistance to including this. A logical step in examining uptake of a new vaccine is comparing it to other similar vaccines.
We did not mean to give off the impression that we were resistant to the comparison, rather we wanted to focus on the NVIs themselves in this study, and not necessarily have a comparative analysis. But we do recognize the importance of comparing the uptake to well-established vaccines in order for the reader to draw a reference. Thus, we have included information and some comparison to the other vaccines. See lines 227-234.
Reviewer 3 Report (New Reviewer)
Comments and Suggestions for Authors
The reply of the authors is no helpful to the comments. The interesting point of the study is regarding the epidemiological status of the country for the two vaccines during that time period
Comments on the Quality of English LanguageModerate changes
Author Response
The reply of the authors is no helpful to the comments. The interesting point of the study is regarding the epidemiological status of the country for the two vaccines during that time period.
We took into consideration the previous comments, and while we recognize there are a number of tables we believe they provide the necessary information for the study which is why we have kept them in the paper. We have added some additional information in the tables (per this reviewer’s and other reviewers’ comments) in an effort to make them more easily understandable. We also appreciated and took into account the comment about included references in the discussion regarding comparisons with similar studies and referred to some in the discussion. As for the epidemiologic status of the country, after the introduction of the RV vaccine, there had been assessments of the status of rotavirus in the country which saw a dramatic decrease in rotavirus-related hospitalizations in the country. Unfortunately, there was no additional information from the time period about the impact of PCV introduction. This information is included in the introduction. See lines 56-62. We recognize the importance of examining the impact of these vaccinations on the diseases themselves and believe that future studies should include an examination of the impact of these vaccinations on the epidemiology of the related diseases. See lines 324-326.
Round 3
Reviewer 3 Report (New Reviewer)
Comments and Suggestions for Authors
Adding to previous comments it is an epidemiological study regarding the specific vaccines. In my consideration there is no improvement to the re-submission of the manuscript
Comments on the Quality of English LanguageModerate changes in English language
This manuscript is a resubmission of an earlier submission. The following is a list of the peer review reports and author responses from that submission.
Round 1
Reviewer 1 Report
Comments and Suggestions for Authors
The authors are strongly recommended to present the negative impact of COVID on individuals' wellbeing. The following studies are recommended to be cited to improve the quality of the paper:
Leung, W. K., Cheung, M. L., Chang, M. K., Shi, S., Tse, S. Y., & Yusrini, L. (2022). The role of virtual reality interactivity in building tourists’ memorable experiences and post-adoption intentions in the COVID-19 era. Journal of Hospitality and Tourism Technology, 13(3), 481-499.
Ågerfalk, P. J., Conboy, K., & Myers, M. D. (2020). Information systems in the age of pandemics: COVID-19 and beyond. European Journal of Information Systems, 29(3), 203-207.
Comments on the Quality of English LanguageModerate english editing is needed.
Author Response
The authors are strongly recommended to present the negative impact of COVID on individuals' wellbeing. The following studies are recommended to be cited to improve the quality of the paper:
Leung, W. K., Cheung, M. L., Chang, M. K., Shi, S., Tse, S. Y., & Yusrini, L. (2022). The role of virtual reality interactivity in building tourists’ memorable experiences and post-adoption intentions in the COVID-19 era. Journal of Hospitality and Tourism Technology, 13(3), 481-499.
Ågerfalk, P. J., Conboy, K., & Myers, M. D. (2020). Information systems in the age of pandemics: COVID-19 and beyond. European Journal of Information Systems, 29(3), 203-207.
Commentary on COVID-19’s impact on wellbeing as well as how it has illuminated the importance and nuances of vaccine uptake is included in the manuscript on L46-51.
Reviewer 2 Report
Comments and Suggestions for Authors
The manuscript is very poorly written. Even the title indicated the author's casual writing skills. Children is written as Chilren. I recommend resubmission of the manuscript after carefully rechecking of the manuscript. References are not in the format of the journal.
Comments on the Quality of English Language
The manuscript is very poorly written. Even the title indicated the author's casual writing skills. Children is written as Chilren. I recommend resubmission of the manuscript after carefully rechecking of the manuscript. References are not in the format of the journal.
Author Response
The manuscript is very poorly written. Even the title indicated the author's casual writing skills. Children is written as Chilren. I recommend resubmission of the manuscript after carefully rechecking of the manuscript. References are not in the format of the journal.
The manuscript was thoroughly reviewed and revised. Please see the revised manuscript.
Reviewer 3 Report
Comments and Suggestions for Authors
Thank you for sharing your article that investigates factors associated with rotavirus and pneumococcus vaccination among Armenian children. Here some comments and suggested edits that could help to improve the article.
L66: Please elaborate more in your manuscript on the 2-stage sampling applied.
L66-67: What is your targeted sample size? Please state in your manuscript the sample size calculation and its assumptions.
L67: Please clarify in your manuscript why you enrolled women aged 15-49 years of age. Were those the mothers of the children investigated?
L72: Did you also perform a sample size calculation of your so called analytic sample of living children or did you simply attempt to investigate all children based on the ADHS? Did parents/caretakers of participating children consent? Please clarify in your manuscript.
L78: How were the households selected? Please state in your manuscript.
L98-100: Did you perform your own PCA? If so, please state in your manuscript how your performed it and which variables you included in your PCA.
Comments on the Quality of English Language
See above.
Author Response
Thank you for sharing your article that investigates factors associated with rotavirus and pneumococcus vaccination among Armenian children. Here some comments and suggested edits that could help to improve the article.
L66: Please elaborate more in your manuscript on the 2-stage sampling applied.
This manuscript is based on a secondary data analysis of an existing de-identified data set from the Armenian Demographic and Health Survey (ADHS). The Demographic Health Survey was conducted by the National Statistical Services and the Ministry of Health of Armenia in collaboration with MEASURE DHS, a USAID-sponsored project. They carried out all stages of survey planning, implementation and data collection. Information on the sampling strategy is included in the manuscript on L73-77.
L66-67: What is your targeted sample size? Please state in your manuscript the sample size calculation and its assumptions.
The ADHS’s overall targeted sample size was calculated by the National Statistical Services of Armenia and the Ministry of Health of Armenia. The targeted sample size was calculated considering the 2010 ADHS. Per the ADHS documentation, “In the 2010 ADHS, household response rates were 87 percent in urban areas and 92 percent in rural areas. The average number of women age 15-49 per household was 0.86 in urban areas and 1.02 in rural areas. Women’s individual response rates were 97 percent in urban areas and 99 percent in rural areas. The sample allocation for the 2015-16 ADHS involved a power allocation with small adjustments because of the large regional size variations. … the 2015-16 ADHS was expected to achieve about 7,000 completed interviews with women age 15-49, with a minimum sample of 514 for the region of Vayots Dzor and a maximum sample of 976 for the city of Yerevan.” All individuals who consented to the survey were included in this analysis. An a priori power calculation was not conducted for this analysis. We do not believe it is appropriate to include a post hoc power calculation in the manuscript, as it would not be done with a priori assumptions, and it is not possible to make an unbiased power calculation once the results have already been visualized.
L67: Please clarify in your manuscript why you enrolled women aged 15-49 years of age. Were those the mothers of the children investigated?
Per the ADHS protocol, all women 15-49 from the selected households were eligible for the survey. These women served as the base for information on children’s immunization status and they were questioned on parity and health of the children in the household. This information is clarified in the manuscript on L77-80.
L72: Did you also perform a sample size calculation of your so called analytic sample of living children or did you simply attempt to investigate all children based on the ADHS? Did parents/caretakers of participating children consent? Please clarify in your manuscript.
All children whose mothers reported vaccination status were included in the analysis, or in our “analytic sample”. As mentioned previously, an a priori power calculation was not conducted for this analysis. We do not believe it is appropriate to include a post hoc power calculation in the manuscript, as it would not be done with a priori assumptions, and it is not possible to make an unbiased power calculation once the results have already been visualized. Included in the manuscript on L83-84.
L78: How were the households selected? Please state in your manuscript.
We have revised the description of household selection to be clearer. All households that were included in the ADHS were part of this analysis. Clarified in the manuscript on L88-89.
L98-100: Did you perform your own PCA? If so, please state in your manuscript how your performed it and which variables you included in your PCA.
The National Statistical Services and the Ministry of Health of Armenia in collaboration with MEASURE DHS performed the PCA and created the wealth indices. This predetermined wealth index variable was used in the analysis and for the sake of clarity of results were grouped into low, middle and high.
Round 2
Reviewer 2 Report
Comments and Suggestions for Authors
This manuscript is not appropriate for publication in the journal Vaccine. The authors may send it to some other journals.
Comments on the Quality of English LanguageThis manuscript is not appropriate for publication in the journal Vaccine. The authors may send it to some other journals.
Reviewer 3 Report
Comments and Suggestions for Authors
Thank you for sharing the revised manuscript. All my comments were addresses sufficiently.
Comments on the Quality of English LanguageSee above.